# Investigation of Microstructure and Properties of Ultra-High Strength Steel in Aero-Engine Components following Heat Treatment and Deformation Processes

Ranhao Yin, Dong Liu, Jingqing Chen and Jianguo Wang *

School of Materials Science and Engineering of Northwestern Polytechnical University, Xi'an 710072, China; yinranhao@mail.nwpu.edu.cn (R.Y.)
* Correspondence: jianguow@nwpu.edu.cn

**Abstract:** C250 steel, renowned for its remarkable strength and toughness, is extensively utilized in the aerospace industry for manufacturing critical components. This research investigates the microstructure and properties of forgings produced through different heat treatment temperatures, aging durations, and thermal cycling intervals. The results demonstrate that the samples were compressed at 1050 °C followed by air-cooling using the conventional maraging treatment. For the cycle heat treatment, temperature was maintained at 1050 °C and cycled 1–2 times, with a heat preservation period of 1 h and subsequent water-cooling. Solution heat treatment at a temperature of 1050 °C, aging for 5 h, and then air hardening were performed to achieve the best forging hardness. Interestingly, the solution time under age hardening conditions had no significant effect on the grain size but had a significant effect on the hardness of martensitic aged steel.

**Keywords:** C250 steel; heat treatment; simulation

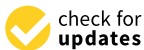



## 1. Introduction

The fan shaft and low-pressure turbine shaft play crucial roles as key components in aero-engines. With their significant size and multi-stage variable cross-sections, these shafts demand exceptional strength and toughness. Maraging steels offer enhanced properties through the precipitation of intermetallic compounds within an ultra-low carbon nickel martensite matrix [1–3]. These steels exhibit ultra-high yield strength, excellent mechanical properties, and superior ductility [4,5] making them widely applied in aviation, aerospace, and navigation industries. Examples of their applications include gyroscope flexible joint [6,7], engine shaft [8,9], rocket engine housings [10], landing gears [11–13], and hulls [14]. However, the machining process often leads to non-uniform microstructure in the narrow strain regions, resulting in severe mixed crystal phenomena. This leads to significant reductions in strength, toughness, and fatigue resistance.

Traditionally, engines have been manufactured using superalloy [15–18]. However, with technological advancements and engine model updates, the conventional processing methods have fallen short in meeting the requirements of modern high-performance aero-engines in terms of cost-effectiveness and performance. This has prompted the exploration of new materials and technologies. Drawing from the successful experience of developed regions such as Europe and United States [19], the utilization of C250 ultra-high strength steel, as a replacement for superalloys, presents several advantages. It not only reduces raw materials and processing costs but also exhibits significantly improved mechanical properties compared to superalloys. Furthermore, the incorporation of C250 steel can contribute to weight reduction in the engine, aligning with the development direction of high-performance aircraft engines. Numerous scholars have proposed various auxiliary analysis methods in this regard. Okkonidis et al. [20] studied the relationship between the structure and mechanical properties of C250 steel under different aging conditions.

Wang et al. [21] established the constitutive equation of 18Ni (250) maraging steel. Ren et al. [22] developed a heat treatment diagram for 18Ni (250) and explored the evolution of its microstructure. Castro et al. [23] examined the microstructure changes in hot forged maraging steel C300. Figueiredo et al. [24–26] explored the microstructure and properties of forged 18NiC300 maraging steel at different solution annealing temperatures, providing a comprehensive characterization of the microstructure parameters across the entire forged sample. However, previous research on hot working of maraging steel has primarily focused on constructing rheological curves and hot working diagrams, with limited attention given to establishing hot working windows. Additionally, the existing fundamental studies on materials have encountered challenges in providing practical guidance for the formulation of production processes. Therefore, there is an urgent need to establish a hot working window for maraging steel to guide actual production process.

In this study, the grain growth behavior of 18Ni(C250) martensitic aging steel was investigated by conducting an empty firing experiment under various heating temperatures and holding times. To examine the microstructure evolution of this ultra-high strength steel under different thermodynamic parameters, namely temperature and strain, an upsetting experiment was performed using a double cone specimen to assess the extent of microstructure recrystallization. The appropriate heat treatment process was subsequently determined through a cyclic heat treatment involving forged and solution aging heat treatments. The obtained results were analyzed and processed, enabling the determination of the recrystallization rate and grain range of 18Ni(C250) maraging steel. Furthermore, a hot working process window was defined to clarify the influence of hot deformation process parameters on the structure and properties of the material, ultimately establishing an optimal control range for the hot deformation process. The influence of the heat treatment process on the structure and properties of maraging steel was elucidated, leading to improved structural uniformity and enhanced integrity of the forgings.

## 2. Materials and Methods

### 2.1. Materials and Experimental Procedures

#### 2.1.1. Materials

The Beijing Iron and Steel Research Institute provided the ultra-high strength maraging steel C250 used in this experiment. Its main chemical composition is shown in Table A1. The yield strength is 1870 MPa, tensile strength is 1945 MPa, elongation is 9%, area shrinkage is 57%, and the maximum reduction rate is 54%. The original structure of the bar is shown in Figure 1, the grain distribution is relatively uniform, and the grain size grade is ASTM 7.

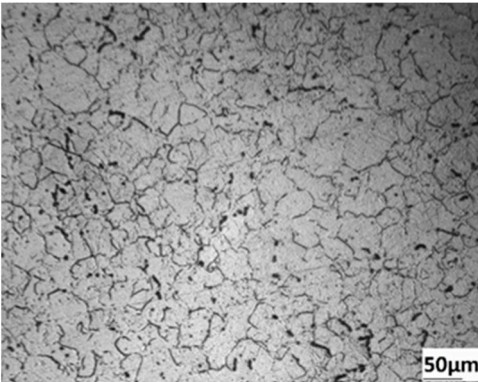

**Figure 1.** The original C250 steel.

#### 2.1.2. Instruments

Thermalization experiments were performed several times without forging in an SX3 ceramic fiber energy-efficient, high-temperature box resistive furnace. The forging process of the C250 steel biconical samples was carried out in a heating furnace and in a 315 T

hydraulic press. The microstructures and grains of the C250 steel after corrosion were photographed by optical microscopy. The Vickers hardness of the C250 steel before and after heat treatment was measured using the MHVD-100IS microhardness tester.

### 2.2. Heat Treatment

The material underwent pre-treated for forging, and a series of experiments were designed to heat the material without forging. Employing the control variable method, the heating temperature in the non-forging experiments varied between 900 and 1050 °C, with a temperature interval of 50 °C. For each temperature, 5 different holding times were selected: 0.5, 1, 1.5, 2, and 4 h, followed by water-hardening. Subsequently, the material was extruded and forged into two different shapes: cylindrical and biconical. Axial stress measurements were conducted to assess the distribution uniformity of strain forces, thereby selecting the optimal forging shape. The cooled samples were divided into 12 equal parts, with 3 parts allocated for forging, cyclic heat treatment, solution aging, and sample preparation. Initially, the effect of experimental temperature and holding time on the average grain size growth of the forgings was investigated using a control variable approach. Subsequently, an empty firing experimental window was established to identify the optimal heating temperature and holding time for the forgings. The degree of recrystallization was then analyzed at different temperatures and stresses, utilizing equally divided pie-shaped samples obtained from forging. Based on the degree of recrystallization, the forging temperature with the most favorable outcomes was selected. Consequently, a cyclic heat treatment was performed to analyze the microstructural grain size shift and identify the optimal heating temperature for the microstructure at different temperatures and cycle times. Finally, the solution method was used to analyze the effect of Vickers hardness and grain size of the forging.

To gain comprehensive insights into the influence of various process parameters on the hot working state of 18Ni(C250) maraging steel, an equivalent diagram was generated using Origin 9.0 software. This diagram presents the hot working window of C250 steel, allowing for a visual representation of tissue changes under different heating conditions. The hot working process window was designed based on the percentage of recrystallized grains and the grain range observed after cyclic heat treatment. This approach facilitated the identification of optimal hot working parameters for C250 steel, ensuring uniform forging structure and desirable outcomes following heat treatment.

### 2.3. Effect of Experimental Temperature and Service Year on Impact Toughness of C250 Steel

Inspired by the work of those who came before [27] and in order to study the characteristics of impact toughness of C250 material, the effects of forging ratio and service time on impact properties of C250 steel were studied. We compared the impact toughness of two structural states of C250 steel under unforged, low forged, medium forged, and high forged (Figure 2A). It can be seen that the impact toughness of C250 steel increases with the increase in forging ratio, and the impact toughness of a high forging ratio sample is close to 50 J cm$^{-2}$. It can be seen from Figure 2B that the impact toughness of emergency reserve steel is much higher than that of steel after long-term service. In both cases, the same batch of metal was tested. This means that the difference in impact toughness is only caused by the microstructure modification that occurs during mining.

The fracture toughness of materials has been studied by predecessors [28]. Studies have shown that with the increase in loading rate, the fracture toughness of the material decreases significantly; that is, the super-strong steel 18NiC250 shows a rate-dependent fracture toughness: when the loading rate increases from $10^{-1}$ MPa·m$^{1/2}$/s to $10^6$ MPa·m$^{1/2}$/s, the fracture toughness decreases by 44 MPa·m$^{1/2}$. For different materials and different fracture forms, the rate correlation shows different forms.

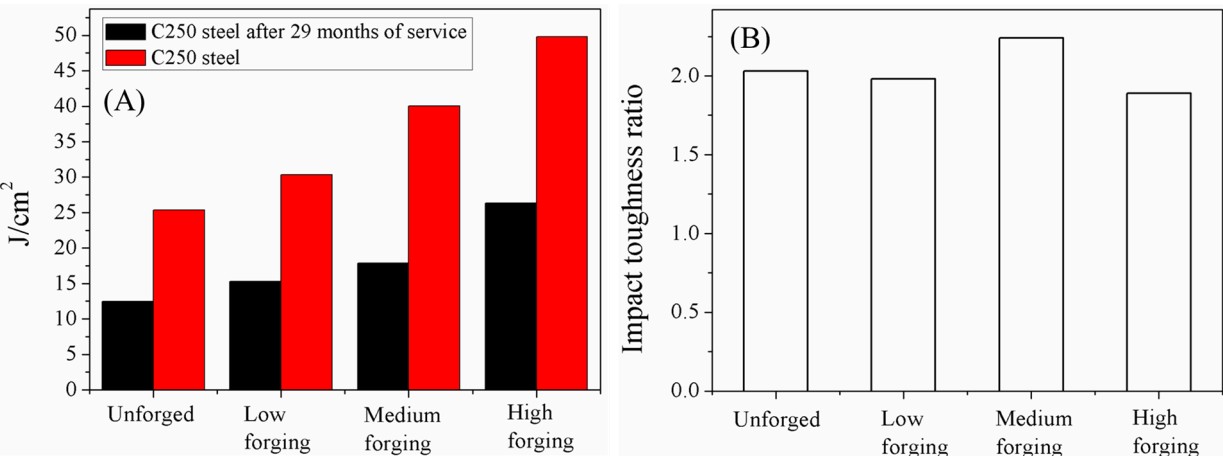

**Figure 2.** Impact toughness value (**A**) and relative change in impact toughness of C250 steel (**B**).

## 3. Experimental Results and Discussion

### 3.1. Selection of Double Cone Specimen

To streamline the experiment and minimize the number of trials, proper design of the shape and size of the C250 steel sample allows for the observation of microstructures in different strain regions at the same temperature. To highlight the improved strain distribution of the biconical specimen after compression, a conventional cylindrical specimen was selected for comparison in the experiment. Utilizing Deform-2D software, the strain distribution of the two specimens was simulated (Figure 3). From Figure 3A, it can be observed that the minimum deformation of the cylindrical sample during the entire deformation process reaches 0.46, while the minimum strain on the edge of the cylindrical sample after deformation is 0.525. The maximum strain does not occur in the deformation core. This is attributed to the initial deformation of the edge, which enhances the metal flow in the core region and reduces deformation resistance. Therefore, if one aims to achieve minimal edge deformation and significant core deformation, it is necessary to delay the onset of edge deformation.

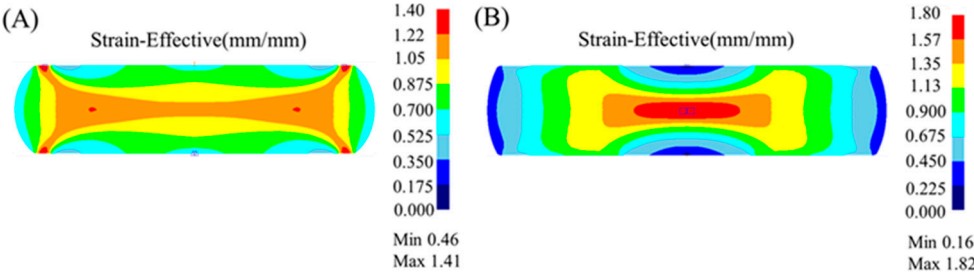

**Figure 3.** Strain field distribution of cylinder (**A**) and biconical specimen (**B**) after deformation.

In response to the failure of the cylindrical sample to meet the experimental requirements, a redesigned sample was developed to address the need for both significant strain in the core and minimal strain at the edge. The specific dimensions of the new sample are illustrated in Figure A3, while the results of the Deform-2D simulation are shown in Figure 3B. By analyzing the strain field distribution of the deformed biconical specimen, it is observed that the edge strain begins at approximately 0.3 and gradually increases from the edge to the core, reaching a maximum strain of approximately 1.8. Compared to the cylindrical samples, the biconical specimen exhibits a wider range of strains, and more importantly, the strains are distributed more uniformly in the radial direction. This allows for enhanced tissue evolution corresponding to a range of different strains. In contrast to the deformation of the cylindrical shape, the deformation process of the biconical shape exhibits preferential deformation in the tapered part, with substantial deformation occurring

at the edge after the deformation of the tapered part is nearing completion. This design approach ensures that the edge experiences minimal deformation, addressing the issue of excessive strain observed in the column-like shape.

In summary, the biconical specimen offers a more uniform strain distribution in the radial direction compared to the cylindrical specimen, effectively avoiding large strain regions at the edges. Consequently, the biconical specimens are more suitable for studying the effect of thermodynamic parameters on the microstructural evolution of C250 steel, as they provide improved experimental conditions.

### 3.2. Effect of Heated Several Times Free from Forging Experiment on Average Grain Size

The surface grains of the forgings were analyzed using a scanning electron microscopy (SEM) of 900, 950, 1000, and 1050 °C. The holding times were set at 1.0, 2.0, and 4.0 h using the control variable method. Figure 4 (for temperatures of 1000 and 1050 °C) and Figure A1 (for temperatures of 900 and 950 °C) illustrate that the average grain size gradually increases with longer holding time while keeping the air firing temperature constant. When the holding time is unchanged, higher air firing temperatures result in larger average grain sizes. Corresponding graphs were constructed to analyze the relationship between average grain size, temperature, and holding time. It is observed that as the heating temperature and holding time increase, the grain size gradually increases, with the influence of temperature being more significant than that of holding time (Figure 5). Below 1000 °C, the effect of holding time on grain growth is not significant. However, at a temperature of 1050 °C, the grain size rapidly increases from 104 μm at a holding time of 0.5 h to 184 μm at a holding time of 4 h. Notably, the change in grain size is more pronounced in the second half of the holding time compared to the first half. A contour map was established to derive the hot working window of the empty firing process, as shown in Figure A2. From this window, it is evident that the optimal heating temperature for the sample is 1050 °C, with a recommended holding time of within 2 h. This heating process holds great significance. These conclusions lay the foundation for subsequent experiments, with the subsequent experimental temperature being set around 1050 °C and the holding time approximately 2 h.

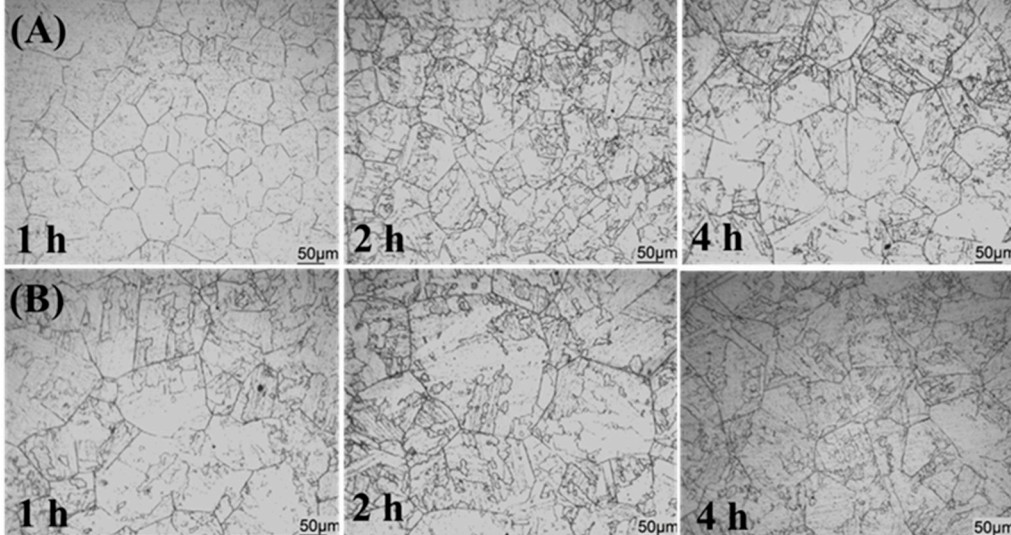

**Figure 4.** The pictures show the microstructure of sintered samples held at (**A**) 1000 °C, and (**B**) 1050 °C for different times.

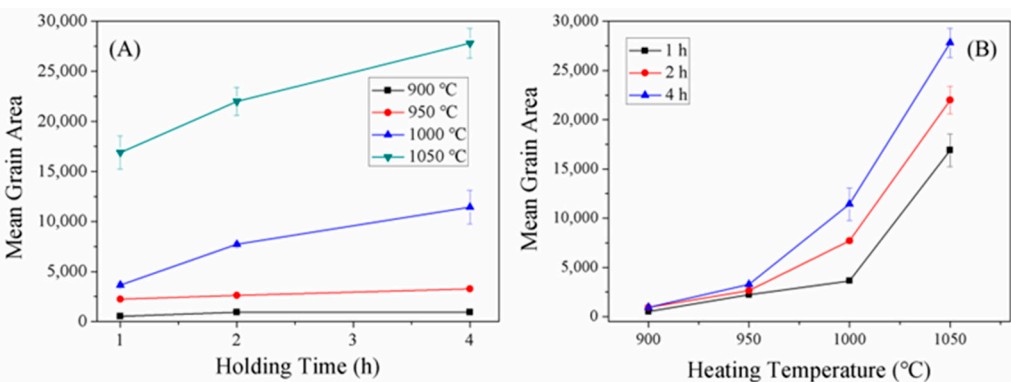

**Figure 5.** Variation curve of the average grain size of C250 steel with (**A**) holding time and (**B**) temperature.

### 3.3. Influence of Forging Treatment on Microstructure and Properties of Maraging C250 Steel

To investigate the impact of forging temperature and equivalent strain on the recrystallization behavior of 18Ni maraging steel (C250), the thermal compression microstructure of a C250 double-cone specimen was characterized by SEM. The results are depicted in Figures 6 and A4. To establish a more direct relationship between strain and the degree of grain recrystallization, a linear correlation diagram was constructed, illustrating the strain and corresponding grain recrystallization rate at different temperatures (Figure 7). When the temperature is kept constant, the recrystallization degree of C250 steel increases with an increase in equivalent strain. Similarly, under a specific strain value, the recrystallization degree of C250 steel rises with elevated temperatures. However, when the temperature exceeds 1050 °C, the degree of recrystallization starts to decrease. Figure 6 also demonstrates the transformation of initial equiaxed crystals in maraging steel to elongated crystals following deformation. Therefore, the heating temperature of C250 steel should be controlled at approximately 1050 °C. In regions characterized by significant deformation, thorough recrystallization of the tissue occurs due to the presence of numerous recrystallized grains surrounding the elongated crystals during formation. This phenomenon contributes to the growth of the recrystallization process.

In terms of the critical strain value at the onset of recrystallization, it is observed that the value gradually decreases as the temperature increases. During the recrystallization process of C250 steel at a specific deformation temperature, it is observed that certain sites with lower strain values did not exhibit any recrystallized grains. This absence of recrystallization can be attributed to the relatively low dislocation density introduced by plastic deformation at these sites. If the critical point is not reached, nucleation does not occur, making recrystallization challenging. However, with an increase in the deformation temperature, the rate of recrystallization is accelerated, leading to the initiation of recrystallization. At a strain value of 0.3, no recrystallized grains were found. Once the strain reaches 0.7 and the deformation temperature exceeds 1100 °C, a small amount of recrystallized grains begins to appear, indicating the initial stages of recrystallization. Nevertheless, when the temperature remains below 1050 °C, no recrystallization was observed.

At a temperature of 1050 °C, the recrystallized grains in the core of the sample are smaller, and the recrystallization degree is higher. As the temperature further increases, the degree of recrystallization continues to increase, but the recrystallized grains also begin to coarsen. This phenomenon can be attributed to excessively high temperatures, which lead to a significantly faster grain growth rate compared to the nucleation rate. Consequently, the recrystallized grains become coarser while simultaneously reducing the critical deformation required for recrystallization. In the temperature range of 900 to 1150 °C, the recrystallization process remains incomplete in the region where the equivalent strain reaches 1.5. The maximum recrystallization percentage is approximately 95%, and there is a noticeable occurrence of mixed crystal in the small strain area. This phenomenon poses a significant challenge for 18Ni steels, as C250 steels exhibit excellent strength and

toughness when the structure is homogeneous. However, when the structure becomes uneven, there is a substantial loss of strength and toughness, resulting in diminished fatigue resistance. Therefore, the application of cyclic heat treatment is continued to improve the uniformity of the structure within the forging.

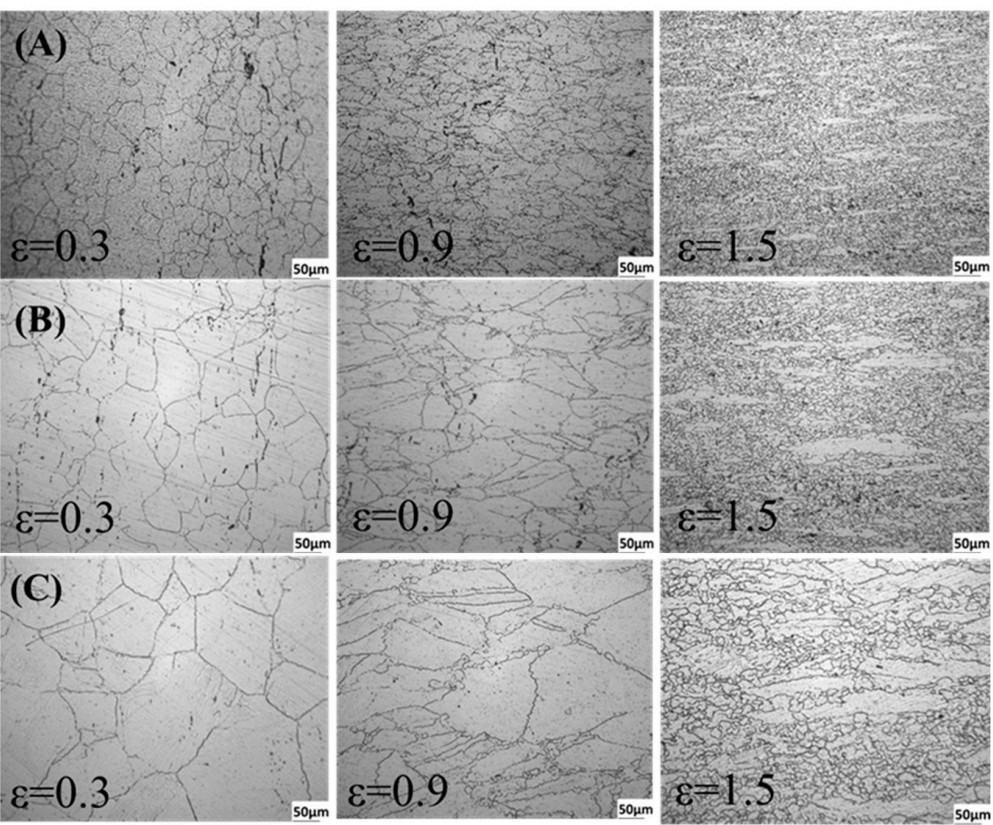

**Figure 6.** Recrystallization behavior of forgings at (**A**) 1000 °C, (**B**) 1050 °C, (**C**) 1100 °C, and different strains.

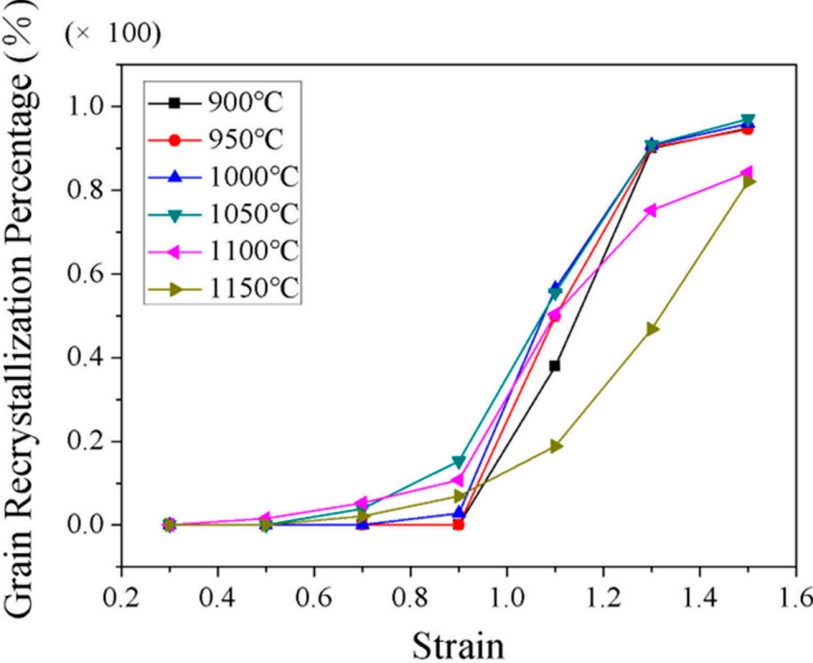

**Figure 7.** The relationship between strain and grain recrystallization rate at different temperatures.

### 3.4. Effect of Cycle Heat Treatment on Microstructure and Properties of Maraging C250 Steel

In the actual production of a low-pressure shaft, different parts of the extrusion experience varying degrees of deformation and recrystallization. Complete recrystallization occurs at large strains, while mixed crystals may appear at small strains. Consequently, cyclic heat treatment is commonly employed after forging to refine the grains and achieve uniformity. The number of cycles of heat treatment varies depending on the material properties, making the investigation of the optimal number of cycles crucial for forging heat treatment studies.

Samples subjected to different forging temperatures and equivalent strain states underwent cyclic heat treatment. The microstructures of the samples forged at 1050 and 1100 °C, following different cycles of heat treatment, are shown in Figure A5. Statistical analysis of the grain sizes in the figure led to the plots depicted in Figure 8, which illustrate the effect of forging temperature and the number of cycles on grain sizes. The combined analysis of the figures reveals that the grain sizes of the samples decreased significantly after three different cycles of heat treatment, regardless of the forging temperature. At 1050 °C, the average grain diameter decreased from 36 μm after one cycle of heat treatment to 19 μm after three cycles of heat treatment, resulting in the refinement of the grains to a grade 8 level. Similarly, at 1100 °C, the average grain diameter decreases from 40 μm after one cycle of heat treatment to 20 μm after three cycles of heat treatment. Cyclic heat treatment not only refines the grain but also improves grain homogeneity. Additionally, it was observed that the sample forged at 1050 °C exhibited the smallest average grain size after cyclic heat treatment. This can be attributed to the higher recrystallization percentage achieved at 1050 °C, as the recrystallization percentage decreased at higher forging temperatures, leading to coarser recrystallized grains.

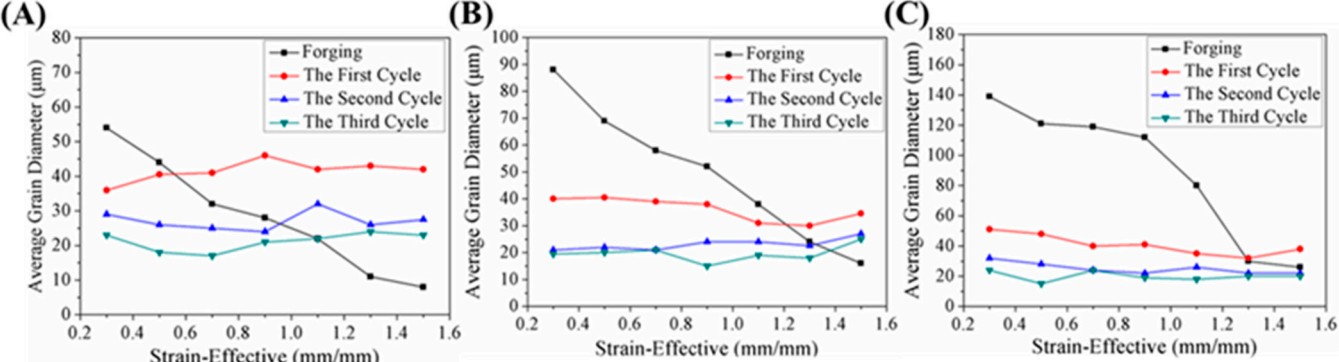

**Figure 8.** The average grain size of forged specimens with different strain values and average grain size after thermal cycling at (**A**) 1000 °C, (**B**) 1050 °C, and (**C**) 1100 °C for different times.

At a constant temperature, the percentage of recrystallization in the material increases as the strain value rises, resulting in grain refinement. Following cyclic heat treatment, grains at small strains undergo further refinement, while grains at large strains grow larger. Furthermore, grains at different strain levels undergo prenucleation and growth, leading to improved grain homogeneity. Even though the average grain diameters vary significantly at different forging temperatures, after three cycles of heat treatment, the average grain diameters concentrate between 16 and 23 μm. The most significant grain refinement occurs during the first-cycle heat treatment, with an average grain size of approximately 6, followed by a range of 7 to 8 for the second-cycle heat treatment. The third-cycle heat treatment shows little difference compared to the second-cycle heat treatment. It is important to note that while cyclic heat treatment can refine the grains of maraging steel, excessive cycles may compromise the material's fatigue resistance. Therefore, it is not always the case that more cycles are better. Depending on the desired grain size and uniformity, one or two cycles of heat treatment may be sufficient.

Based on the aforementioned analysis, it is evident that the recrystallization rate is highest, and the grain size is finer, during hot working at 1050 °C, which aligns with the simulation results in Section 3.2. Therefore, in this study, a heating temperature of 1050 °C is chosen for forging at the same time, and 1–2 cycles of heat treatment are applied since the heat treatment effect is the best. To further improve the strength of the forgings, the solid solution-strengthening method is employed.

### 3.5. Effect of Solid Solution Heat Treatment on Microstructure and Properties of Maraging C250 Steel

The solution treatment of 18Ni(C250) maraging steel involves dissolving elements such as Ni, Co, Mo, and others into the matrix, forming a supersaturated replacement solid solution. This solution treatment enhances the microstructure strength by hindering dislocation movement through distortion [29]. Therefore, the forgings experience a significant increase in strength after undergoing solution treatment.

Observing the microstructure diagram after solution aging (Figure A6), a slight growth trend in grain size can be noticed. However, the aging time has no significant effect on the grain size, which mainly influences the sample size. To provide a clearer representation, the average grain size under different solution aging conditions was mapped, as shown in Figure 9. Examining the mechanical properties (Figure 10), Vickers hardness values do not exhibit significant differences among different forging temperatures, heat treatment systems, or strain positions. The average Vickers hardness value of the samples forged at 1050 °C at different strain positions is 315.9 HV. However, after cyclic heat treatment, the hardness decreased significantly. The Vickers hardness value after one cycle of heat treatment is 280.2 HV, while it is 276.7 HV after the second cycle. After three cycles of heat treatment, the hardness value slightly increases to 294.1 HV, showing a minor difference compared to the second cycle. On average, the Vickers hardness value decreases by approximately 10% in the cyclic heat treatment state. In order to show the influence of different processing methods on the hardness of forging more sensitively, we study it by measuring its microhardness [30]. The results are shown in Table 1. Following solid solution aging, it becomes evident that the Vickers hardness and microhardness values of the sample gradually increase with longer aging time. The average microhardness value at different strain values rises from approximately 320 HV in the forged state to about 550 HV after solid solution aging. This represents an average increase in Vickers hardness values of over 70%.

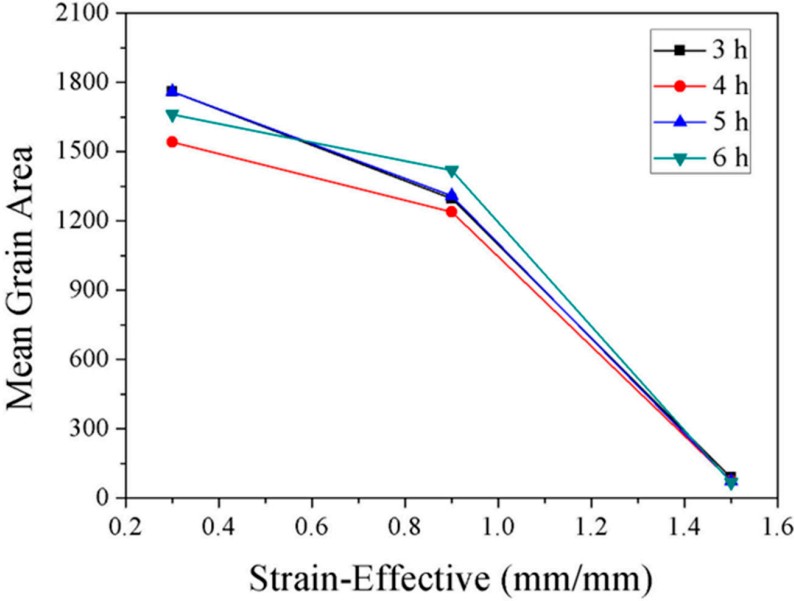

**Figure 9.** The relationship between strain and grain size under different solution aging.

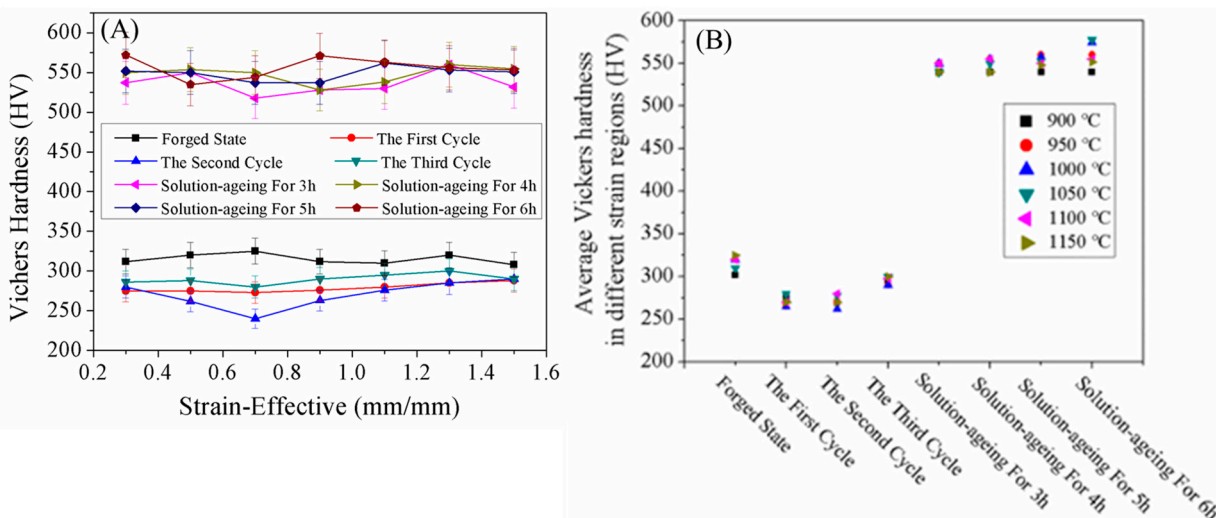

**Figure 10.** The relationship between Vickers hardness and strain and heat treatment state and heat treatment temperature. Under different strain and heat treatment conditions, the average Vickers hardness value of the sample after forging at 1050 °C (**A**). The average Vickers hardness value of forgings under different heat treatment states and temperatures (**B**).

**Table 1.** Influence of different treatment methods on the microhardness of forgings.

| | Strain-Effective (mm/mm) | Hardness (Microhardness) | |
|---|---|---|---|
| | | **H** | **ΔH** |
| **Forged State** | 0.3 | 312 | −3.9 |
| | 0.5 | 320 | 4.1 |
| | 0.7 | 325 | 9.1 |
| | 0.9 | 312 | −3.9 |
| | 1.1 | 310 | −5.9 |
| | 1.3 | 320 | 4.1 |
| | 1.5 | 308 | −7.9 |
| **The First Cycle** | 0.3 | 275 | −40.9 |
| | 0.5 | 275 | −40.9 |
| | 0.7 | 273 | −42.9 |
| | 0.9 | 276 | −39.9 |
| | 1.1 | 280 | −35.9 |
| | 1.3 | 285 | −30.9 |
| | 1.5 | 288 | −27.9 |
| **The Second Cycle** | 0.3 | 280 | −35.9 |
| | 0.5 | 262 | −53.9 |
| | 0.7 | 240 | −75.9 |
| | 0.9 | 263 | −52.9 |
| | 1.1 | 276 | −39.9 |
| | 1.3 | 285 | −30.9 |
| | 1.5 | 290 | −25.9 |
| **The Third Cycle** | 0.3 | 286 | −29.9 |
| | 0.5 | 288 | −27.9 |
| | 0.7 | 280 | −35.9 |
| | 0.9 | 290 | −25.9 |
| | 1.1 | 295 | −20.9 |
| | 1.3 | 300 | −15.9 |
| | 1.5 | 290 | −25.9 |

**Table 1.** *Cont.*

| Strain-Effective (mm/mm) | Hardness (Microhardness) | |
| | H | ΔH |
|---|---|---|
| **Solution-aging for 3 h** | | |
| 0.3 | 537 | 221.1 |
| 0.5 | 550 | 234.1 |
| 0.7 | 518 | 202.1 |
| 0.9 | 528 | 212.1 |
| 1.1 | 530 | 214.1 |
| 1.3 | 560 | 244.1 |
| 1.5 | 532 | 216.1 |
| **Solution-aging for 4 h** | | |
| 0.3 | 550 | 234.1 |
| 0.5 | 554 | 238.1 |
| 0.7 | 550 | 234.1 |
| 0.9 | 528 | 212.1 |
| 1.1 | 538 | 222.1 |
| 1.3 | 560 | 244.1 |
| 1.5 | 555 | 239.1 |
| **Solution-aging for 5 h** | | |
| 0.3 | 552 | 236.1 |
| 0.5 | 550 | 234.1 |
| 0.7 | 537 | 221.1 |
| 0.9 | 537 | 221.1 |
| 1.1 | 562 | 246.1 |
| 1.3 | 553 | 237.1 |
| 1.5 | 551 | 235.1 |
| **Solution-aging for 6 h** | | |
| 0.3 | 572 | 256.1 |
| 0.5 | 535 | 219.1 |
| 0.7 | 544 | 228.1 |
| 0.9 | 571 | 255.1 |
| 1.1 | 563 | 247.1 |
| 1.3 | 556 | 240.1 |
| 1.5 | 553 | 237.1 |

In conclusion, the hardness of the sample slightly reduces after cyclic heat treatment, but it experiences a considerable improvement after solid solution aging.

*3.6. Establishment of a Hot Working Process Window for the Percentage of Recrystallized Grains*

In order to gain a comprehensive understanding of the influence of various process parameters on the hot working state of 18Ni(C250) maraging steel and provide a theoretical basis for the production process of low-pressure turbine shaft extrusions in aviation, Origin 9.0 software was used to draw an iso-map. This iso-map reveals the hot working window of C250 steel, allowing for a visual examination of microstructure changes under different heating conditions. The establishment of the hot working window in this study primarily focuses on two key aspects: the percentage of recrystallized grains and the grain size after cyclic heat treatment. In this section, particular attention is given to establishing the hot working process window based on the percentage recrystallized grains.

Table 2 shows the percentage of recrystallized grains for different forging temperatures and strain values, while Figure A7 depicts the hot working window as a contour plot based on the collected data. The recrystallization percentage window for C250 steel, reveals a general increase in the percentage of recrystallized grains with increasing equivalent strain. The hot working window, as shown in Figure A7, can be divided into three distinct regions from bottom to top: no recrystallization region, mixed crystal region, and fully recrystallized region. In the fully recrystallized region, where the percentage of recrystallized grains exceeds 95%, a homogeneous crystal structure is achieved. To attain this state, a strain value of 1.6 or higher is required at temperatures below 1050 °C, while a strain value of

1.5 or higher is necessary at 1050 °C and above. The bottom region of the hot working window represents a region where the fraction of recrystallized grains is less than 5%. In this region, recrystallization is essentially absent, and the dominant grains are relatively coarse, deformed grains. When the equivalent strain is less than 1.0 at a low temperature or below 0.7 at high temperatures, little to no recrystallization occurs within this region. The middle part of the hot working window consists of a mixed grain structure, where deformed grains coexist with small, recrystallized grains. Based on the findings of the hot working window analysis, to ensure the attainment of a uniform and fine recrystallized structure after forging, it is recommended to prioritize larger strains (above 1.5) and an intermediate temperature (approximately 1050 °C) within the established hot working window.

**Table 2.** The fraction of recrystallized grains of the tissue at different forging temperatures and strain values.

| Equivalent Strain | Temperature (°C) | | | | | |
|:---:|:---:|:---:|:---:|:---:|:---:|:---:|
| | 900 | 950 | 1000 | 1050 | 1100 | 1150 |
| 0.3 | 0 | 0 | 0 | 0 | 0 | 0 |
| 0.5 | 0 | 0 | 0 | 0 | 0.016 | 0.001 |
| 0.7 | 0 | 0 | 0 | 0.039 | 0.052 | 0.024 |
| 0.9 | 0 | 0 | 0.028 | 0.153 | 0.107 | 0.269 |
| 1.1 | 0.306 | 0.598 | 0.564 | 0.555 | 0.643 | 0.688 |
| 1.3 | 0.564 | 0.851 | 0.907 | 0.909 | 0.852 | 0.868 |
| 1.5 | 0.888 | 0.906 | 0.929 | 0.950 | 0.952 | 0.961 |

*3.7. Establishment of a Thermal Processing Window for Grain Range after Cyclic Heat Treatment*

In general, increasing the number of cycles of heat treatment for the forgings leads to smaller average grain sizes and improved structural uniformity. However, it is important to strike a balance, as excessive cycles can lead to a decline in forging properties and fatigue resistance. Therefore, this study aims to minimize the number of cyclic heat treatments while ensuring the desired structural homogeneity. To achieve this, it is necessary to accurately establish the grain size range window of the material at different forging temperatures and strain levels after various cycles of heat treatment. A contour map of the grain size range, as shown in Figure 10, is obtained by analyzing the statistics of grain size ranges in the cyclic heat treatment structure map. Figure 11A illustrates the grain size range after one cycle of heat treatment. At temperatures around 1050 °C, the smallest range of grain sizes is observed for strain values above 0.7, with a range of approximately 45 μm. However, the grain size is still relatively large, indicating the need for further cyclic heat treatment. Figure 11B represents the grain size range after the second cycle of heat treatment. At a temperature around 1050 °C, the range of strain values above 0.8 reaches approximately 30 μm, which meets the requirements for a uniform state. However, at 1100 °C and above, it is observed that the original recrystallized grain size increases significantly, resulting in a broader range of microstructure grains after cycling. Figure 11C depicts the grain size range after three cycles of heat treatment. It is evident that the grain size is greatly reduced across the entire range, but the range of grain sizes in small strain regions at high temperatures remains relatively high.

In summary, even after three cycles of heat treatment, the grain size range in the small strain region remains relatively large for each temperature condition. At temperatures of 1100 °C and above, the significant difference in grain size during the first and second heat treatment cycles with reduced equivalent effect does not meet the requirements of structural homogenization. At 1050 °C, when the equivalent strain value reaches 1.0 or higher, a forging with a uniform structure can be obtained.

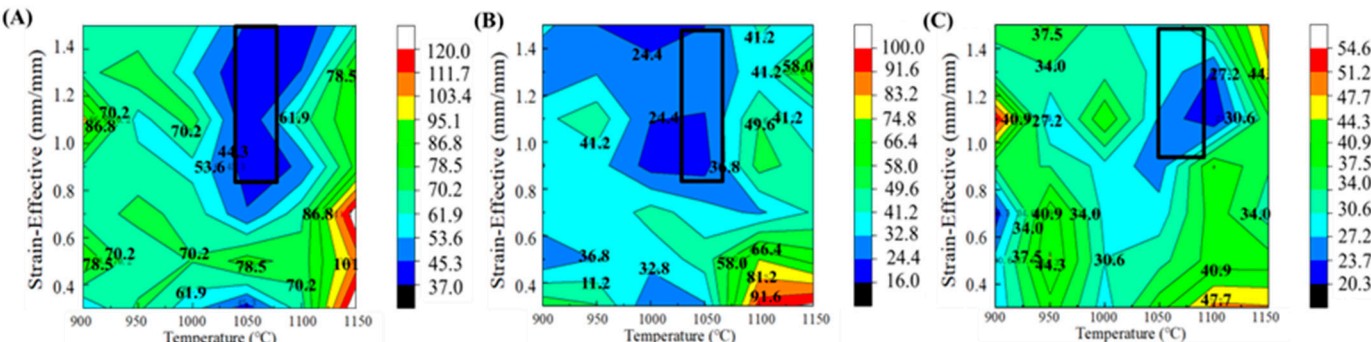

**Figure 11.** Extremely poor grain size after one (**A**), two (**B**), and three (**C**) cycles of heat treatment.

## 4. Conclusions

This study focused on the investigation of 18Ni(C250) martensitic steel through various experiments, including heating without forging, double-cone compression, and post-forging heat treatment. The main objectives were to analyze the percentage of recrystallized grains and the grain size after cyclic heat treatment, and to establish a hot working process window for 18Ni(C250) martensitic steel with poor recrystallization percentage and grain size. The following conclusions can be drawn from this study:

1.  Increasing the heating temperature and prolonging the holding time led to increases in the average grain size. The effect of holding time on grain size is smaller compared to the effect of the temperature. At lower temperatures, the change in grain size with holding time is less severe compared to higher temperatures. Grain growth is more prominent in the first half of the holding time and milder in the second half.
2.  During the hot working of 18Ni maraging steel, it was observed that when the temperature is between 900 and 1150 °C and the equivalent strain is below 1.5, the recrystallization percentage does not reach 100%, and there is a significant presence of mixed crystals in the small strain region. Therefore, optimization of process parameters is recommended in practical production to avoid the strain dead zones. According to the results of the heating without forging experiment and the double-cone experiment, an extrusion temperature of approximately 1050 °C is suggested in actual production.
3.  After one cycle of heat treatment, the grain size in the small strain region can be refined from grade 3 to 6. After three cycles of heat treatment, further refinement to grade 8 is achieved. The effect of refinement is not significantly different between the second and third cycles. Therefore, it is recommended to subject the forged specimens to no more than two cycles of heat treatment, considering the magnitude of the strain.
4.  After the solution aging treatment, there is a slight increase in grain size, while the hardness increases by over 70%. The hardness value gradually increases with the aging time.
5.  Analysis of the grain range window after one cycle of heat treatment indicates that the smallest grain size range is observed at approximately 1050 °C for strain values above 0.7, with a range of about 45 µm. The grain range after the second cycle of heat treatment shows that when the temperature is between 1000 and 1050 °C, the range for strain value above 0.8 is approximately 30 µm, which meets the requirements for a homogenous crystal state. After three cycles of heat treatment, the overall grain size range is significantly reduced. However, the grain size range in the small strain region at high temperatures remains relatively high. Based on the grain range window analysis, it is recommended to select a temperature around 1050 °C during the forging process to achieve a fine and uniform structure, while ensuring an equivalent strain value above 1.0.

**Author Contributions:** Conceptualization, R.Y. and J.C.; methodology, D.L.; software, J.C.; formal analysis, R.Y.; investigation, D.L.; resources, J.W.; data curation, R.Y.; writing—original draft preparation, R.Y.; writing—review and editing, J.W.; visualization, J.C.; supervision, J.W.; project administration, D.L.; funding acquisition, J.W. All authors have read and agreed to the published version of the manuscript.

**Funding:** The authors would like to thank the National Natural Science Foundation of China (No. 52101052) for their support.

**Data Availability Statement:** Data available upon request from the authors.

**Acknowledgments:** Thanks to Shun Han, Chunxu Wang, and Yong Li of the Central Iron and Steel Research Institute for their contributions to this work.

**Conflicts of Interest:** The authors declare no conflict of interest.

## Appendix A

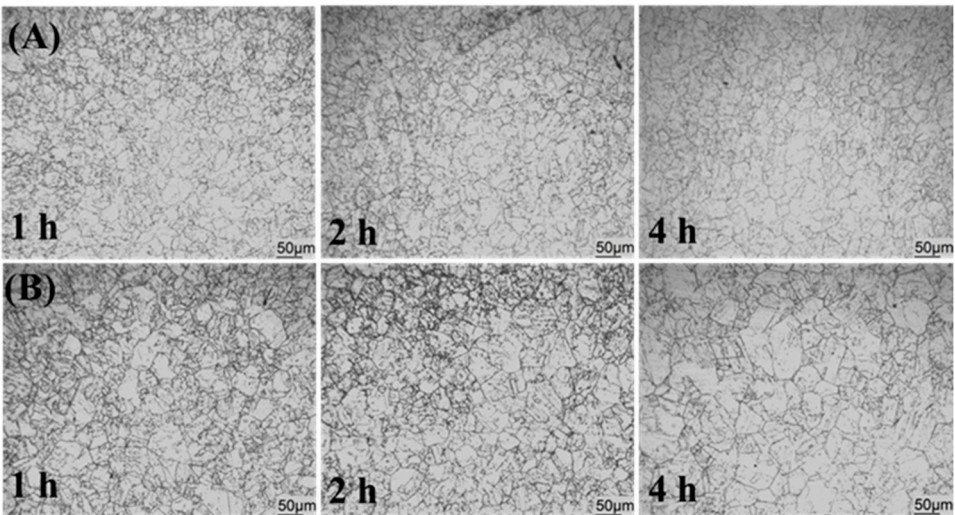

**Figure A1.** The pictures show the microstructure of sintered samples held at (**A**) 900 °C and (**B**) 950 °C for different times.

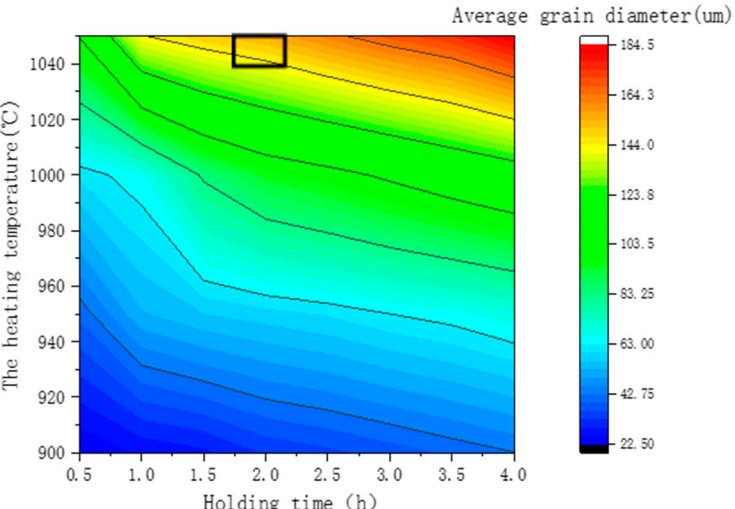

**Figure A2.** Heated several times free from forging experiment window.

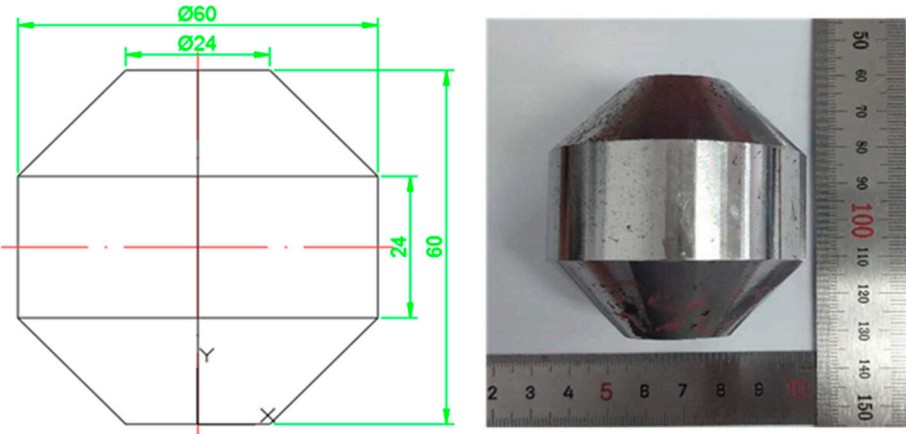

**Figure A3.** Cone shape and size.

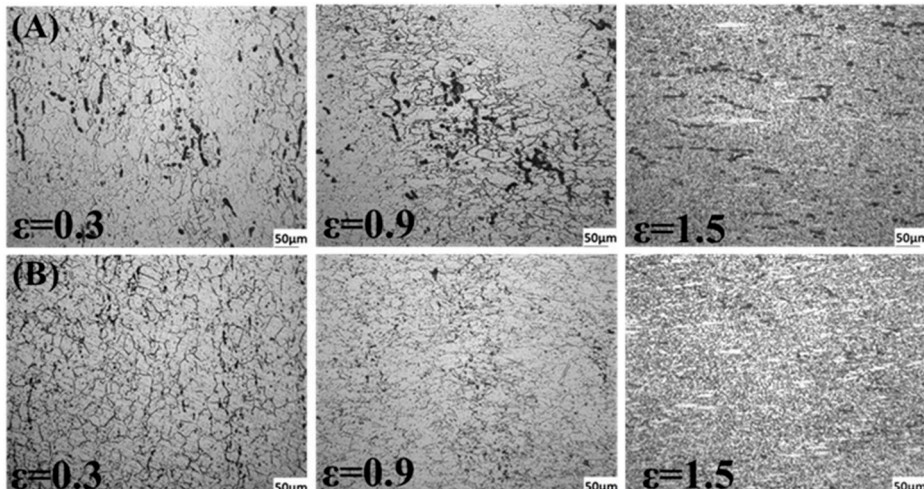

**Figure A4.** Recrystallization behavior of forgings at (**A**) 900 °C and (**B**) 950 °C and different strains.

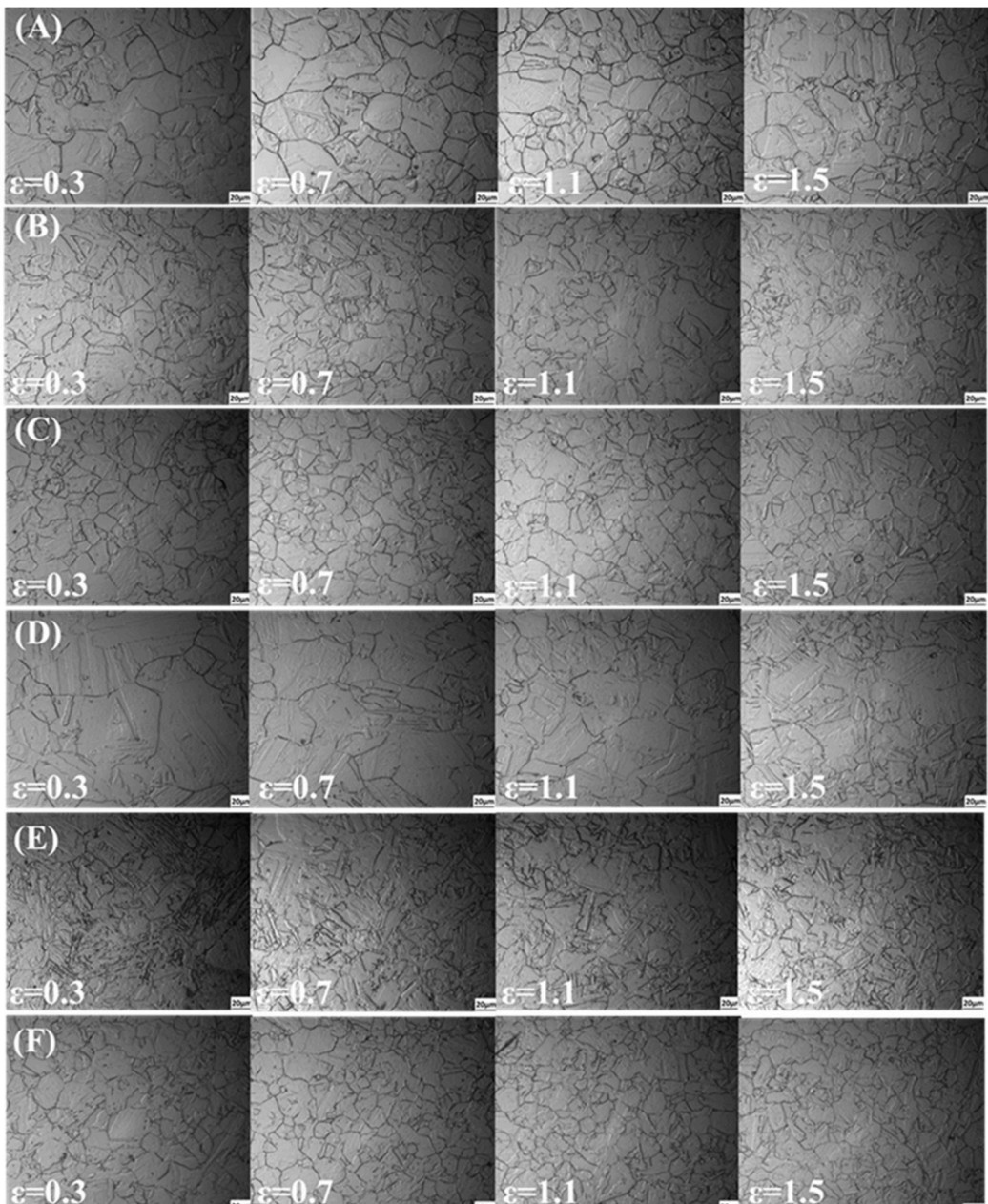

**Figure A5.** Effects of different temperatures of 1000 °C (**A**–**C**) and 1050 °C (**D**–**F**) and different heat treatment times (**A**,**D**) once, (**B**,**E**) twice, and (**C**,**F**) three times on the grain size of forgings.

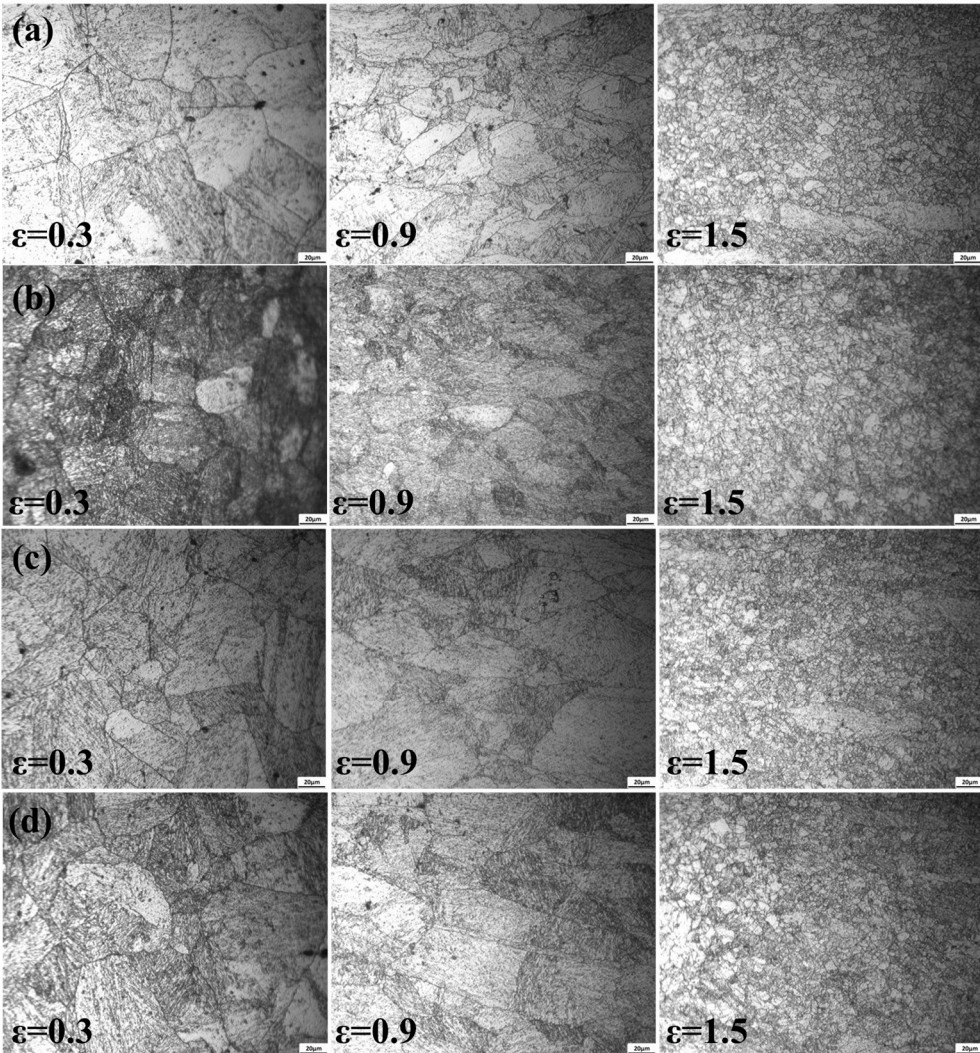

**Figure A6.** The grain structure of forgings after aging treatment for (**a**) 3 h, (**b**) 4 h, (**c**) 5 h, and (**d**) 6 h.

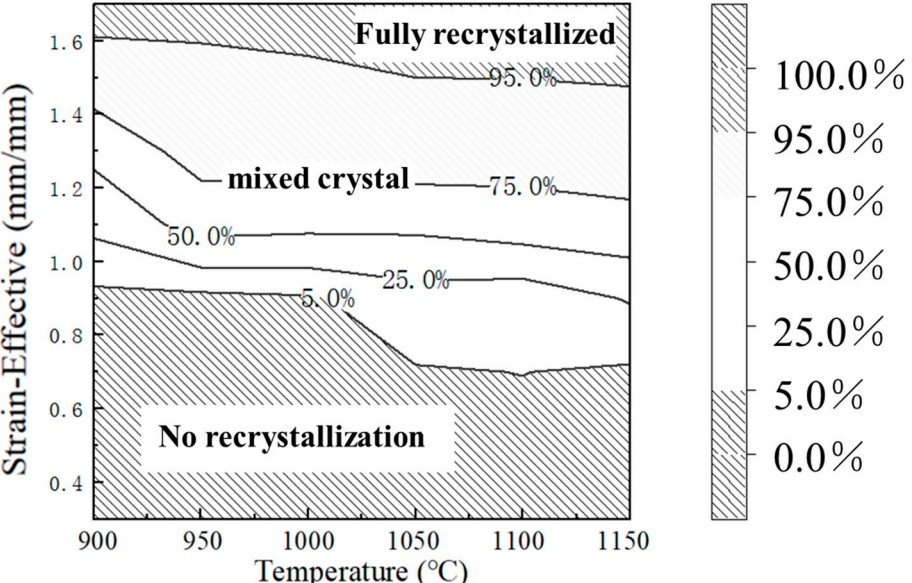

**Figure A7.** Processing window for the percentage of C250 steel recrystallized grains.

**Table A1.** Composition table of C250 maraging steel for experiment.

| Ni | C | Co | Mo | Ti | Al | Mn | P | Si | S |
|---|---|---|---|---|---|---|---|---|---|
| 17.97 | 0.004 | 8.02 | 5.01 | 0.41 | 0.1 | 0.01 | 0.003 | 0.01 | 0.001 |

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
