# Peer review of "Investigation of Microstructure and Properties of Ultra-High Strength Steel in Aero-Engine Components following Heat Treatment and Deformation Processes"

_metals, doi:10.3390/met13081353_

Round 1
Reviewer 1 Report (Previous Reviewer 2)
This is a good systematic study aimed at finding the regimes of heat and deformation-heat treatment of ultra-high strength maraging steel, providing the most uniform fine-grained microstructure. However, a number of points need to be improved before the final publication of the paper.
- The Abstract should be revised, starting with the phrase ‘The results demonstrate…’ Reading the text, the essence of the study is completely incomprehensible. In addition, the last phrase in the Abstract can be misleading, because hardness after aging, on the contrary, increased greatly.
- Lines 29-30. ‘hybrid crystal phenomenon’. Here it is necessary to explain to the reader what is meant.
- Line 77. ‘Strength 1870 MPa’. What strength do the authors mean? Is this the yield strenght?
- Line 84. ‘respectively’. Delete it.
- Specify the dimension on the X-axis of Figure 2. In addition, add markers A and B in Figure 2.
- Specify the dimension on the X-axis of Figure 7.
- Add markers A and B in Figure 8 and their description in the figure caption.
- Lines 318-321. The end of the phrase doesn't seem to contain a verb.
- Lines 329 and 348. Better 'strain values' instead of 'strain rates'
- Line 341. ‘low?’ Why the question mark ?
Author Response
Please see the attachment.

Reviewer 2 Report (New Reviewer)
The authors of the work study the change in the grain size of C250 maraging steel after various thermo-mechanical treatments. The article is technological, contains the technological recommendations and may have a high practical interest for engineers.

Author Response
Please see the attachment.

Reviewer 3 Report (New Reviewer)
A significant reduction in grain size, as well as the formation of a developed substructure in the internal volumes (grain boundary and substructural hardening), in fact, is the only hardening mechanism that simultaneously increases both the resistance to plastic deformation and the resistance to brittle fracture. The study of these processes will allow a more reasonable approach to the choice of modes of strengthening heat treatment of structural steels. The article obtained valuable practical and scientific results, but there are several questions:
1. On fig. 8, the authors used hardness as a parameter for assessing the state of the material, but perhaps it is insensitive. Maybe it would be better to use microhardness, it is more sensitive to grain size changes and morphological transformations in alloys, look for example: https://link.springer.com/article/10.1007/s11003-008-9077-z
2. The article does not evaluate the parameters of plasticity, in particular, the relative narrowing of the samples. This does not allow a full assessment of the plasticity of the material.
3. The article would become more informative if a formula or algorithm for choosing the optimal parameters of thermo-mechanical processing was proposed.
4. The article did not determine the characteristics of crack resistance and impact strength, but these parameters are very important for steel:
https://www.sciencedirect.com/science/article/pii/S1877705817301170
5. Was the statistical processing of the obtained mechanical parameters carried out and by what methods?
Round 2
Reviewer 1 Report (Previous Reviewer 2)
Missing (B) in Figure 10 caption. Please fix it in the final version paper. Everything else is acceptable.
Author Response
Thank you
Reviewer 3 Report (New Reviewer)
Accept.
Author Response
Thank you very much! Thank you for your help with my manuscript!
This manuscript is a resubmission of an earlier submission. The following is a list of the peer review reports and author responses from that submission.
Round 1
Reviewer 1 Report
Please revise the work and rewrite the manuscript in an accessible scientific language to capture the readers’ interest. Your work might be good and interesting, but the language is currently the barrier to figure this out, hindering the understanding of the content. I also noticed that your manuscript heavily depends on SEM micrographs (“pictures” in the manuscript!); however, to the reader, they appear indistinguishable and inconclusive. I would rethink the thesis of the manuscript and present only the data/results that support the argument. If you find something interesting in SEM micrographs, please annotate and make it easy for the reader to find it too.
My final recommendation is to look at previously published articles at MDPI Metals on a similar topic to see how your manuscript’s quality compare to them.
Author Response
Point 1: Please revise the work and rewrite the manuscript in an accessible scientific language to capture the readers’ interest.
Response 1: Thank you very much for your comments. We have checked and revised the language of the article in detail.
Point 2: I also noticed that your manuscript heavily depends on SEM micrographs (“pictures” in the manuscript!); however, to the reader, they appear indistinguishable and inconclusive. I would rethink the thesis of the manuscript and present only the data/results that support the argument. If you find something interesting in SEM micrographs, please annotate and make it easy for the reader to find it too.
Response 2: Thanks for the reminder. We have added a description of the phenomenon that the image in the article wants to represent.
Point 3: My recommendation is to look at previously published articles at MDPI Metals on a similar topic.
Response 3: Thank you very much for your advice. I read MDPI Metals' previous articles, revised the inaccurate expression in the article, and specifically modified the format of the reference and the problem of unclear numbers in the pictures.

Reviewer 2 Report
This is a good and systematic study on the effect of heat and deformation treatment parameters on the grain structure characteristics of C250 maraging steel. The article is of great practical importance, because the authors set the parameters of heat and deformation treatments, providing the most uniform and fine-grained steel structure. However, the article requires significant revision and, as presented, cannot be recommended for publication.
1) A serious concern is the English in the paper. A very substantial re-editing is required before it can be considered for publication:
- there are typos, e.g. 'riginal' instead of 'original' (line 413); 'Figure S6' (line 308)
- there is misuse of terms, e.g. 'stresses' instead of 'strains' (line 199); 'percentage' instead of 'fraction' (line 323), ‘tissue’
- there are bad constructions, for example 'processing process' (line 24), 'average grain size of the grains' (lines 232-233)
- there is an incorrect construction of the phrase, for example, ‘The traditional engine processing technology is made of superalloy’ (line 28), ‘Then set the air-burning experiment window, and select the best heating temperature and holding time’ (lines 92-93)
- there are incorrect/unclear phrases, e.g. ‘From the equivalent strain at the same temperature, with the increase of the strain value, recrystallization When the percentage increases, the grains begin to be refined’ (lines 238-240); ‘the hot pier coarsening experiment’
- phrase repeat (line 354)
- and etc.
2) The list of references should be expanded; in addition, it must be formatted in accordance with the guidelines of the journal.
3) The numbers in Figure 8 are not visible.
4) The title of the manuscript should be corrected as was not only heat treatment, but deformation processing too.
Author Response
Point 1: A serious concern is the English in the paper. A very substantial re-editing is required before it can be considered for publication:
- there are typos, e.g. 'riginal' instead of 'original' (line 413); 'Figure S6' (line 308)
- there is misuse of terms, e.g. 'stresses' instead of 'strains' (line 199); 'percentage' instead of 'fraction' (line 323), ‘tissue’
- there are bad constructions, for example 'processing process' (line 24), 'average grain size of the grains' (lines 232-233)
- there is an incorrect construction of the phrase, for example, ‘The traditional engine processing technology is made of superalloy’ (line 28), ‘Then set the air-burning experiment window, and select the best heating temperature and holding time’ (lines 92-93)
- there are incorrect/unclear phrases, e.g. ‘From the equivalent strain at the same temperature, with the increase of the strain value, recrystallization When the percentage increases, the grains begin to be refined’ (lines 238-240); ‘the hot pier coarsening experiment’
- phrase repeat (line 354)
- and etc.
Response 1: Thank you very much for your advice. We've made changes to the entire article. Fixed spelling errors, terminology abuse, and sentence structure issues in the article.
Point 2: The list of references should be expanded; in addition, it must be formatted in accordance with the guidelines of the journal.
Response 2: Thanks for the reminder. We've added articles from recent years and revised the reference format.
Point 3: The numbers in Figure 8 are not visible.
Response 3: Thanks for the suggestion. We have enlarged and bolded the numbers in Figure 8.
Point 4: The title of the manuscript should be corrected as was not only heat treatment, but deformation processing too.
Response 4: Thank you very much for your suggestions. We refined the manuscript title.

Round 2
Reviewer 1 Report
The initial comments, unfortunately, have not addressed well enough.
Author Response
Response:
Dear Professor,
We have revised and confirmed the language of the article and the reference of professional terms. And check the logic and expression of the whole article, the picture mentioned in the article to do a detailed description, linking the preceding and the following, so as to facilitate the reader's reading.
Thank you very much for your valuable suggestions on this work!

Reviewer 2 Report
Although English has been improved, further polishing is needed.
Author Response
Response:
Dear Professor,
We have revised and confirmed the language of the article and the reference of professional terms. And check the logic and expression of the whole article, so as to facilitate the reader's reading.
Thank you very much for your valuable suggestions on this work!

Round 3
Reviewer 1 Report
None at this time.